# *CardioRVAR*: A New R Package and Shiny Application for the Evaluation of Closed-Loop Cardiovascular Interactions

**DOI:** 10.3390/biology12111438

**Published:** 2023-11-16

**Authors:** Alvaro Chao-Écija, Manuel Víctor López-González, Marc Stefan Dawid-Milner

**Affiliations:** 1Autonomic Nervous System Unit, CIMES, School of Medicine, University of Málaga, 29071 Malaga, Spain; alvarochao@uma.es (A.C.-É.); manuelvictor@uma.es (M.V.L.-G.); 2Biomedical Research Institute of Málaga (IBIMA), 29590 Malaga, Spain

**Keywords:** autonomic nervous system, baroreflex sensitivity, vector autoregressive modeling, discrete wavelet transform, R programming environment

## Abstract

**Simple Summary:**

The main objective of this work was to integrate a review of methodologies to estimate closed-loop baroreflex sensitivity, an index used for autonomic nervous system evaluations, into a new software package, *CardioRVAR*, developed for the R programming environment. We present the results obtained after applying the tool in different scenarios and discuss its potential applications in a clinical setting. The tool is open-source and stored in a public GitHub repository, which allows feedback and provides room for the improvement and continuation of the work.

**Abstract:**

*CardioRVAR* is a new R package designed for the complete evaluation of closed-loop cardiovascular interactions and baroreflex sensitivity estimated from continuous non-invasive heart rate and blood pressure recordings. In this work, we highlight the importance of this software tool in the context of human cardiovascular and autonomic neurophysiology. A summary of the main algorithms that *CardioRVAR* uses are reviewed, and the workflow of this package is also discussed. We present the results obtained from this tool after its application in three clinical settings. These results support the potential clinical and scientific applications of this tool. The open-source tool can be downloaded from a public GitHub repository, as well as its specific *Shiny* application, *CardioRVARapp*. The open-source nature of the tool may benefit the future continuation of this work.

## 1. Introduction

Non-invasive methods to study autonomic functions are common and highly popular in the field of autonomic neurophysiology. We would like to highlight in this context the contributions of Rodríguez-Liñares et al. [1] and Martínez et al. [2] to the R programming environment in the form of the RHRV, a software package designed for the assessment of the state of vagal and sympathetic modulation in individuals through the evaluation of heart rate variability (HRV), estimated from interbeat intervals (IBI) and heart rate (HR) recordings [1,2,3]. In this work, the focus of study comprises both the evaluation of said HRV together with the behavior of the mechanisms that control both HRV and blood pressure variability (BPV).

There are several procedures to estimate HRV and BPV. The procedure of interest in this work is the frequency-domain assessment method, which consists of the identification and isolation of the activity present in two frequency bands associated with adrenergic and vagal modulation: respectively, the low-frequency (LF) band, consisting of frequencies from 0.04 to 0.15 Hz, and the high-frequency (HF) band, which encompasses frequencies from 0.15 to 0.4 Hz [3,4]. These components are extracted from HR and blood pressure (BP) recordings, such as systolic or diastolic blood pressure (SBP, DBP). One should note that, in BPV, this HF band would be representative of the respiratory activity, whose frequency components are usually located at this band [3,4].

In the frequency domain, the magnitude that relates both BPV and HRV is the spontaneous baroreflex sensitivity (BRS) [4]. The BRS can be estimated as the transfer function [4,5,6] between both measurements, which is usually computed as the ratio between the cross-spectrum of the IBI series and the BP recording, and the BP auto-spectrum [5,6], and it is summarized in Equation (1) as described in works such as Barbieri et al. [5] (p. 36) or Faes et al. [6] (p. 278):(1)$$\alpha_{o}\left(f\right)=\frac{S_{BP,IBI}\left(f\right)}{S_{BP,BP}\left(f\right)}$$

However, this method assumes an open-loop model to describe the dynamics of the cardiovascular system [5]. A closed-loop model can be used to take into account both feedforward and feedback interactions, in which HR and BP both influence themselves by different mechanisms: either mechanical ones that induce changes in BPV through changes in HRV, or baroreflex ones that induce changes in the opposite direction [5,7]. Moreover, these variabilities may be influenced by other factors external to this closed-loop, which should also be considered [5,7,8].

In this context, several toolboxes that consider this type of model have been designed for MATLAB, most of them based on contributions from Faes et al. [9,10,11], for the analysis of EEG signals and closed-loop cardiovascular data. From these MATLAB tools, we highlight the *eMVAR* toolbox [9] (http://www.lucafaes.net/emvar.html (accessed on 30 August 2023)) and the *blockMVAR* toolbox (http://www.lucafaes.net/blockmvar.html (accessed on 30 August 2023)), designed for the evaluation of causal relationships between variables through autoregressive modeling [8,9,12]. It is also worth mentioning the works of Barnett and Seth in the form of the *GCCA* [13,14] and *MVGC* [15] MATLAB toolboxes, which, although designed for and centered on neural connectivity analysis, share both procedures and core algorithms with the ones employed to analyze closed-loop cardiovascular data.

In the R statistical environment, the authors have recently proposed a software tool for the determination of spontaneous BRS changes using wavelet analyses [16]. In addition, we highlight R-based package *grangers* (https://github.com/MatFar88/grangers (accessed on 30 August 2023)), by Farnè and Montanari and based on their work [17], which allows causal analysis between pairs of signals. However, the R environment lacks sufficiently specific and detailed contributions such as the ones designed for MATLAB for the measurement of baroreflex sensitivity using causal closed-loop models of interactions.

We present in this work the R package *CardioRVAR*, a tool developed in the R environment to better describe these interactions in individual subjects by working at the same time with both the present feedback and feedforward mechanisms that regulate the studied variables, using the R statistical environment for data analysis. In the following sections of this work, a review of the methodology integrated into the package will be presented, as well as the general workflow of the package.

## 2. Materials and Methods

### 2.1. Vector Autoregressive Models

The core of this work is centered on the computation of vector autoregressive (VAR) models [5,6,8,9,12,13,14,15,17,18,19]. These models are able to describe causal interactions between lagged values of the modeled variables against themselves as well as against the other involved variables, until a specific lag conventionally denoted as ***p*** is reached, a limit that corresponds to the order of the VAR model [5,6,8,9,12,13,14,15,17,18,19]. Equation (2), as in Barnett and Seth [15] (pp. 52–53) or Faes et al. [6] (p. 278), describes the general structure of a bivariate **VAR(*p*)** model [5,17], modified as in Faes et al. [9] (p. 4) to include ***a*****(0)** coefficients in the structure:(2)$$\begin{array}{c} y\left[n\right]=\sum_{k=0}^{p}a_{11}\left(k\right)y\left[n-k\right]+\sum_{k=0}^{p}a_{12}\left(k\right)x\left[n-k\right]+w_{y}\left[n\right]\\ x\left[n\right]=\sum_{k=0}^{p}a_{21}\left(k\right)y\left[n-k\right]+\sum_{k=0}^{p}a_{22}\left(k\right)x\left[n-k\right]+w_{x}\left[n\right] \end{array}$$

If this model were to be applied to a system describing the interactions between variables ***x*** and ***y***, the first branch of Equation (2) would represent the main process of this system (e.g., how HRV influences BPV) whereas the second would describe a feedback controller for the main process (e.g., the baroreflex interaction between BPV and HRV) [5,8]. One should note that the traditional VAR model only considers lagged interactions; therefore, all direct and instantaneous interactions would be due to the model residuals, and ***a*_11_(0) = *a*_12_(0) = *a*_21_(0) = *a*_22_(0) = 0** [8,19]. By applying the Fourier transform on the autoregressive coefficients as described in Equation (3) [5] (p. 35) [9] (p. 4), one can obtain the frequency-domain representation of these coefficients, and, thus, of the whole model, as presented in matrix notation in Equation (4) [5] (p. 35):(3)$$A\left(f\right)=\sum_{k=0}^{p}a\left(k\right)e^{-i2\pi fk}\;$$
(4)$$\left[\begin{array}{c} Y\left(f\right)\\ X\left(f\right) \end{array}\right]=\left[\begin{array}{cc} A_{11}(f) & A_{12}(f)\\ A_{21}(f) & A_{22}(f) \end{array}\right]\left[\begin{array}{c} Y\left(f\right)\\ X\left(f\right) \end{array}\right]+\left[\begin{array}{c} W_{Y}\left(f\right)\\ W_{X}\left(f\right) \end{array}\right]\;$$

From Equation (4), one can derive the functions that constitute the closed-loop transfer functions for each branch of the system. Equations (5) and (6) represent, respectively, the transfer functions of the main process and the feedback controller described in Equation (2), as discussed in Barbieri et al. [5] (p. 35):(5)$$\beta_{c}\left(f\right)=\frac{A_{12}\left(f\right)}{1-A_{11}\left(f\right)}\;$$
(6)$$\alpha_{c}\left(f\right)=\frac{A_{21}\left(f\right)}{1-A_{22}\left(f\right)}\;$$

However, this model, in the context of cardiovascular variables, is actually biased [7,8,9]. In order to correct it, direct transfer paths not previously included should be allowed from one of the variables to the other [7,8,9,19]. By rearranging Equation (4), one can arrive at the following matrix equation, given by Equation (7) as in Barbieri et al. [5] (p. 36) or Barnett and Seth [15] (p. 52):(7)$$S=H\Sigma H^{*}\;$$

Here, ***S*** is a matrix representing the frequency-domain cross-spectral matrix of the previous variables ***X*** and ***Y***, whereas ***Σ*** represents the noise-covariance matrix of the model, and ***H*** is a matrix containing the transfer functions that link the activity of the estimated noise sources of the model to the modeled variables [5,8,13,15,19]. In addition, operator ***** applied to matrix ***H*** in Equation (7) represents its conjugate transpose [8,13,15,19]. Matrix ***Σ*** will therefore contain every direct instantaneous interaction in the model, as the main model is designed so that its coefficients represent the captured lagged interactions [8]. These instantaneous interactions can be isolated and incorporated into the main structure of the model by estimating a matrix of interactions capable of diagonalizing the original matrix *Σ* [8,9,19]. This process is described in Equation (8) [8] (p. 104):(8)$$\Sigma =D\Sigma_{diag}D^{T}$$

Then, the coefficients of matrix ***D*** will be incorporated into matrix ***H*** and will be propagated back to the transfer functions previously described in Equations (5) and (6) [8]. This makes either ***a*_12_(0) ≠ 0** or ***a*_21_(0) ≠ 0**, depending on which direct transfer path is selected, as these paths are unidirectional, while generating a new noise-covariance matrix [8,9,19]. Not only are the corresponding transfer functions estimated, but also one can evaluate the specific contribution that each noise makes to each variable, which can be indicative of the causal strength in the interactions among the modeled variables [8]. These computations and transformations are applied to the frequency-domain representation of the model, but they could also be performed on the time-domain model [9,19]. The time-domain representation of this type of MVAR model is defined by Faes et al. [9,19] as its extended version and also represents the core of the *eMVAR* MATLAB toolbox.

### 2.2. Causal Coherence and Noise Contribution

We integrated in our package the computation of the causal coherence measure, based on the works of Porta et al. [18] and Faes et al. [6,9,19]. As explained in the literature [6,9,18,19], the causal coherence of a bivariate series refers to the value of the spectral coherence after suppressing one of the possible causal interactions, i.e., turning to zero a specific portion of the matrix of interactions of the modeled system, representative of a certain interaction. Therefore, the spectral coherence and the causal coherence can be defined using Equations (9) and (10), respectively, as reported in Porta’s work [18] (p. 243):(9)$${{coh}^{2}}_{SBP,IBI}\left(f\right)=\frac{{\left|S_{SBP,IBI}\left(f\right)\right|}^{2}}{S_{SBP,SBP}\left(f\right)S_{IBI,IBI}\left(f\right)}$$
(10)$${{coh}^{2}}_{SBP\to IBI}\left(f\right)={{{coh}^{2}}_{SBP,IBI}\left(f\right)|}_{H_{SBP,IBI}(f)=0}$$

Causal coherence therefore would measure the coupling strength of a single interaction. The measurement of causal coherence can also be used as an indicator of baroreflex mechanisms [6,18]. A related measure to the causal coherence is the noise source contribution measure, which is also related to causal interactions [8]. As defined in the literature, the noise source contribution is calculated as the percentage of a certain spectrum that is explained by a particular noise source [8]. In a bivariate process, such as an IBI-SBP model, the IBI spectrum is influenced by both the IBI white noise source and, to a certain degree, the SBP noise source [8].

### 2.3. Trend Removal with the Discrete Wavelet Transform

In addition, a brief summary of the discrete wavelet transform [20] (DWT) is given here, as it has been considered for the preprocessing algorithms of this software. The DWT is able to decompose a signal into detail [20,21] and approximation or smooth coefficients [20,21], each contained in a specific decomposition level representing a frequency band, whose limits are defined by powers of two [20,22]. The frequency ranges contained in each band are determined by the number of decomposition levels applied, and the so-called Nyquist frequency (i.e., half of the sample frequency of a given signal) [20,22]. In this work, this decomposition is performed with the maximal overlap discrete wavelet transform (MODWT) [21]. An example of the result of this type of decomposition algorithm on a given signal and its potential to isolate and detrend said signal is depicted in Figure 1 and Figure 2, in which decompositions were performed using the Haar [22,23] wavelet. In Figure 2, an additional border handling strategy [24] is added to the decomposition, therefore producing a different representation of the coefficients.

As one can observe in Figure 1, due to the applied number of decomposition levels, the approximation coefficients form a representation of a very-low-frequency trend of the signal, while the detail coefficients contain information for each analyzed frequency band, represented by a specific decomposition level. This allows the design of an algorithm capable of isolating such trends, which consists of editing the coefficients calculated from the decomposition process by comparing each coefficient to a certain threshold [25,26]. After the coefficients have been edited, an inverse transform is performed to estimate and then subtract a trend from the original signal, resulting in a detrended version of said signal [25,26]. While other detrending methods, such as the smoothness priors [25,26,27], could also be performed, the wavelet detrending method achieves the desired results in a more computationally efficient manner [25,26]. The importance of this detrending process will be discussed in the following sections.

### 2.4. CardioRVAR Workflow

The discussed tool is currently stored in two GitHub repositories. The first one, *CardioRVAR*, allows the user to work with an R package and to use the provided functions directly in the R console. The second one, *CardioRVARapp*, houses a *Shiny* application (https://shiny.posit.co/ (accessed on 15 October 2023)) [28] that can automate the available routines of *CardioRVAR* through a graphical user interface. *CardioRVAR* operates on the algorithms previously described to achieve a complete profile of the cardiovascular interactions present among HR and BP variables. The software algorithm can be summarized into the following steps:Select a data file with *CardioRVARapp* and upload it into the software structure;Resample the uploaded time series after selecting a certain frequency, if needed;Manually select from the estimated HR and BP recordings a specific time window of interest;Preprocess the windowed data, first by subtracting the mean of each series and then by detrending them with an adaptative MODWT-based algorithm, and validate them accordingly [13,14] for further analyses;Estimate a time-domain **VAR(*p*)** model from the validated chosen segments and then diagnose and validate the model with the criteria described in the literature [13,14,15,29];Transform the model into the frequency domain;Extract instantaneous unidirectional interactions from this frequency-domain representation, given a specific zero-lag-interaction path already chosen by the user, and update the model with such interactions;Estimate the most important features of the model and then display and report them;Generate and report single-subject indices from the model, allowing the user to choose a method to estimate said indices.

This workflow can be achieved using the functions provided in the *CardioRVAR* package, or automatically by using the graphical user interface *CardioRVARapp*. The workflow will be discussed in more detail in the following subsections, and transcripts of the required commands from the current version of *CardioRVAR* will be provided.

### 2.5. Data Upload and Preprocessing

After uploading a data file, several functions can be used to preprocess said data. The first preprocessing steps would be to interpolate and detrend the data, as well as to assess their stationarity, as indicated in the literature [1,13,14,15,29,30]. This can be achieved by using the following commands.
*> library(CardioRVAR)**> # Data is a list with elements Time, RR, and SBP:**> Data <- ResampleData(Data, 4) # Interpolates data**> IBI <- DetrendWithCutoff(Data$RR) # Detrends IBI signal**> SBP <- DetrendWithCutoff(Data$SBP) # Detrends SBP signal**> New_Data <- cbind(SBP = SBP, RR = IBI)**> CheckStationarity(New_Data) # Checks stationarity of the data**[1] TRUE**> # Or alternatively:**> Check_stationarity <- CheckStationarity(New_Data, verbose = TRUE)**Time series are stationary*

*CardioRVAR*’s graphical interface allows one to upload data from csv and txt files, which contain information regarding beat-to-beat BP and IBIs. During the program testing, these csv files were obtained from the *ACQKnowledge* v4.2.0 software. The application also offers the possibility to interpolate each recording, if not done yet, to a sample frequency of 4 Hz by default, as suggested in the literature [1,30]. After uploading the data, a visual representation of the HR tachogram associated with the SBP data will be displayed on-screen. The user will now have the possibility to select a time window to isolate a specific interval from the recording for its analysis. Mean HR and SBP values of the isolated segments will be reported once the window is selected. Figure 3 depicts this interface and this process.

Once a time window has been chosen, the windowed signals are then preprocessed by the R package, which results in detrended segments of the original signals through the MODWT algorithm, adjusted through a border handling strategy (Figure 2) by default. The detrending process is necessary for two main reasons: first, the detrending method has been designed to mitigate the effects of the very-low-frequency (VLF) band [3,27], as shown in Figure 4; second, the detrending allows one to make the data stationary, which is a necessary step in order to further analyze the data [13,14]. As with the smoothness priors detrending method, for which a frequency cutoff can be selected to better perform the detrending [25,27], the nature of the MODWT allows one to define a frequency cutoff, or a reference frequency from which the cutoff can be computed, that will determine which frequency bands will be contained in the estimated trend [25].

Based on the strategy suggested by Li et al. [25], the detrending method included in *CardioRVAR* adapts the number of decomposition levels to the sample frequency of the series, in combination with a desired cutoff or reference frequency provided by the user. For a specific decomposition level ***d***, a frequency cutoff ***f_C_*** would be defined by Equation (11), as the allowed frequency bands’ limits are powers of two and depend on the Nyquist frequency ***f_N_*** [20,22,25]:(11)$${f_{C}}_{d}=\frac{f_{N}}{2^{d}}$$

If a reference frequency is set by the user instead of a proper cutoff, the cutoff will be computed as the lower limit of the frequency band to which the reference frequency belongs. In other words, if a frequency ***f_ref_*** is selected, *CardioRVAR* will find a decomposition level associated with an interval of possible reference frequencies $\left[\left.{f_{C}}_{d},{f_{C}}_{d-1}\right)\right.$ that contains ***f_ref_*** and will use said interval to select frequency ${f_{C}}_{d}$ as the cutoff. By reordering Equation (11), and adapting it to include reference frequencies, we can obtain Equation (12), which integrates the described steps and generates the optimal decomposition level according to a reference frequency lower than the Nyquist frequency, with $\left\lceil \cdot \right\rceil$ representing the ceiling operator:(12)$$d=\left\lceil \log_{2}f_{N}-\log_{2}f_{ref}\right\rceil$$

For a reference frequency of 0.04 Hz, as in Li et al. [25], and a sample frequency of 4 Hz, the detrending algorithm will generate a six-level decomposition, in which the last decomposition level will contain a frequency band between 0.03125 Hz and 0.0625 Hz. From this band, its lower limit, which is also the upper limit of the frequency band represented by the approximation coefficients, will be used as a cutoff. *CardioRVAR* assumes by default a sample frequency of 4 Hz and uses a reference frequency from the interval [0.03125 Hz, 0.0625 Hz) (e.g., 0.04 Hz, the lower limit of the LF band), and therefore the frequency 0.03125 Hz, as suggested by Li et al. [25], as a cutoff to estimate the trend, reducing the VLF component of the signal (Figure 4B).

This cutoff can be modified by the user so as to ensure the stationarity of the detrended signals: the lower the frequency cutoff, the more the VLF component is retained, while this component will be more mitigated if the cutoff is increased (Figure 4C). By selecting, for example, a reference frequency from the interval [0.0625 Hz, 0.125 Hz) (e.g., 0.07 Hz), a five-level decomposition will be performed, and the frequency 0.0625 Hz will be used as a cutoff. Using a high cutoff or reference frequency, however, may also have the undesirable effect of affecting the spectral components of interest, especially the LF component, as shown in Figure 4C.

Moreover, the use of a specific wavelet to perform the detrending is important as, apart from mitigating the VLF components, it is desirable to preserve the HF and LF components as much as possible. An example of the importance of this choice is shown in Figure 5, where the relative errors between the non-parametric spectra of a raw IBI signal and its detrended version using the MODWT-based algorithm included in *CardioRVAR* are depicted. Two different wavelets were used to perform the detrending, the Haar and the Daubechies 8 [23] wavelets, which are commonly renamed in R packages such as *RHRV* as *haar* and *d16*, respectively [23]. While both wavelets generate high relative errors below 0.04 Hz, indicating that the VLF component has been mitigated, the Daubechies 8 wavelet tends to maintain lower relative errors for the HF and LF components than the Haar wavelet. This indicates that the former is better for preserving these HF and LF components and therefore is more suitable for the detrending process. *CardioRVAR* allows the use of several wavelets to modify the detrending and operates with the Daubechies 8 wavelet by default.

After this detrending is performed on the windowed data, *CardioRVAR* then tests each segment to check if they are stationary. As suggested by Seth’s work [13,14] on the use of MVAR models to describe and analyze neurological connectivity, the augmented Dickey–Fuller (ADF) and the KPSS tests are employed for this task. Once the detrended segments are statistically confirmed to be stationary, *CardioRVAR* starts the data modeling.

### 2.6. Analysis of Cardiovascular Closed-Loop Interactions

Once the data have been preprocessed, *CardioRVAR* proceeds to generate a **VAR(*p*)** model of the chosen segments, which will be defined by a specific model order or maximum lag limit and will capture the interactions of interest present at said segments. This model order is usually chosen by applying an information criterion, which, in the case of *CardioRVAR* is, by default, the Akaike Information Criterion (AIC) [31,32], as suggested by authors such as Faes et al. [9] regarding closed-loop cardiovascular analysis. However, the software allows one to select the model order that users consider more appropriate. It should be noted that very low model orders will decrease the resolution of the frequency-domain results, and too high model orders will produce extra peaks in the variability spectra that may deviate from the true frequency-domain representation of the studied signals [32].

Once the model is estimated, certain statistical criteria are used to validate it. These criteria are widely reported in the literature and consist of assessing the stability of the model and the white noise nature of the model residuals [13,14,15,29]. While the validation criteria used by *CardioRVAR* were proposed in the context of neural connectivity analysis, the systems that are analyzed in said context share similar properties with the ones analyzed in a cardiovascular one, such as the possible presence of cross-correlated noise sources in the systems [8,13]. This justifies the use of said criteria for cardiovascular analyses. *CardioRVAR* reports both these features, as well as the best model order according to the chosen information criterion, the AIC by default, for a specific multivariate signal. When a valid model has been estimated, one can use it as a source to calculate frequency-related data. *CardioRVARapp* performs this routine automatically using its graphical interface; if the user wants to do this by themselves using the functions provided by *CardioRVAR*, the following are the commands that should be used.
*> # Data represents a matrix with two interpolated time series, IBI and SBP.**> Data[,“IBI”] = DetrendWithCutoff(Data[,“IBI”])**> Data[,“SBP”] = DetrendWithCutoff(Data[,“SBP”])**> # Both signals have been detrended with these commands.**> CheckStationarity(Data)**[1] TRUE**> # A VAR model is estimated from the stationary time series and then validated:**> model <- EstimateVAR(Data)**> Check_residuals <- DiagnoseResiduals(model, verbose = TRUE)**Model residuals are white noise processes**> Check_stability <- DiagnoseStability(model, verbose = TRUE)**The model is stable*

With these commands, once the data have been checked for stationarity, a model can be estimated from the data and can also be checked to ensure its validity. The estimated model can then be used to further assess its properties in the frequency domain.

### 2.7. Analysis in the Frequency Domain

The main functionality of *CardioRVAR* is to transform the time-domain **VAR(*p*)** models into their frequency-domain representations. To do so, one should use the following command, which will be applied to the previous *model* object:
*> freq_model <- ParamFreqModel(model).*

When calculating this new representation, the software estimates both branches of the cardiovascular closed-loop model and allows users to choose which branch of the model should be evaluated by selecting a specific input and output. If SBP (HR) is the input (output), baroreflex interactions will be evaluated. On the contrary, mechanical effects will be evaluated if HR (SBP) is chosen as the input (output).

*CardioRVAR* allows users to choose a direct transfer path from one of the variables into the other. Although users are free to choose which direct transfer path they wish to include in the model, from SBP to HR or vice versa, we strongly suggest the former option, as the literature considers this a means of including fast baroreflex interactions in the model [6,8]. For this reason, *CardioRVAR* also reports these instantaneous interactions after isolating them from the frequency-domain representation of the model.

### 2.8. Assessment of the Transfer Functions

*CardioRVARapp* is able to display three types of transfer functions (Figure 6). The first one is a biased representation of the closed-loop transfer function, as it is calculated without taking into account the chosen direct transfer paths and is only shown for comparison purposes. The second one is the corrected closed-loop transfer function, adjusted to allow these direct transfer paths. The third one is the classical open-loop transfer function, obtained either from Equation (1) or as the *type I* index indicated by Barbieri et al. [5] (p. 36). All the representations are accompanied by their respective indices to allow statistical comparisons. The interpretation of these transfer functions depends on the evaluated arm of the loop: for example, Figure 6 depicts the calculated transfer functions with SBP (HR) as the main input (output) of the arm, with a direct transfer path from SBP to HR, which is also reported by the software (not shown in the figure). This transfer function represents the baroreflex sensitivity of the analyzed data.

Several methods have been introduced in *CardioRVAR* to estimate expected values from the computed transfer functions. The first method is to perform this estimation by simply reporting the arithmetic mean of the transfer function at each frequency band of interest. This method, however, is not recommended as it does not evaluate the reliability of the estimates and could induce misrepresented expected values [33,34]. Thus, other methods were introduced in the package to tackle this problem. The second one is the classical thresholding method, which makes use of the squared spectral coherence to estimate the reliability of the data by only considering estimates whose associated coherence values surpass a certain threshold, which is usually 0.5 n.u. [8,33]. This method, however, can be too strict to the point of sometimes not obtaining valid estimates from a particular subject due to low coherence values [34,35]. The method proposed by McLoone and Ringwood [35] was also introduced in our software, which consists of applying a frequency-domain Gaussian window to the transfer function at a specific band before computing the expected values. As both authors point out, with this method, the center of each frequency band is mainly considered for the evaluation of the baroreflex activity [35]. Finally, another estimation method, which consists of reporting the value of the transfer function at the maximum coherence value of the bands of interest [36], is also included in our software.

### 2.9. Assessment of the Noise Source Contribution and Causal Coherence

As the computed models consider noise sources, one can evaluate their contribution to the variability of the analyzed segment. This is reported by *CardioRVAR* in two different ways: first, the software reports the percentage of this contribution as defined by Hytti et al. [8], from each noise source to each type of variability (Figure 7); second, it can be achieved as a frequency-domain causal coherence function. This can also be interpreted as a causality measure: the more that one type of variability can be explained by external noise sources concerning a certain variable, the stronger these causal interactions [8]. Figure 7 depicts an example of a noise source evaluation obtained from a patient before and during a head-up tilt (HUT) test, which was displayed by *CardioRVARapp*.

### 2.10. Evaluation of the Tool: Data Sources

In this work, we present three analyses conducted with the tool in order to test its effectiveness. The first analysis consists of the descriptive study of different autonomic and baroreflex features from two subjects: a healthy subject and a patient with Postural Orthostatic Tachycardia Syndrome (POTS) induced by COVID-19, the latter of which is a phenomenon that has been recently reported in the literature [37]. The second analysis consists of a comparison of estimates produced from the tool from five individual normotensive subjects and seven hypertensive subjects in the context of HUT sessions. In the final analysis, estimates computed with the tool from subjects from the EUROBAVAR data set [34] are compared in the supine and standing positions, a validation strategy already used in other works [16,35,38,39]. For these two last analyses, the normality of distributions in the former analysis and of the difference in distributions in the latter is tested with the Shapiro–Wilk test. Then, depending on whether normality can be rejected or not, a Wilcoxon test or a *t*-test for unpaired samples, in the case of the former analysis, and for paired samples, in the case of the EUROBAVAR data, is performed to compare the estimates, assuming a confidence level of 0.95. More information regarding the composition of this data set, which includes two baroreflex-impaired subjects, is available in the work of Laude et al. [34].

## 3. Results and Discussion

### 3.1. Descriptive Study of Two Subjects

We offer a practical example in which our software was used to evaluate cardiovascular interactions from certain subjects, who will be referred to as subjects A and B. Both subjects participated in a HUT test, during which HR and beat-to-beat non-invasive SBP recordings were obtained. In each example, three intervals were considered for the analysis, which will be referred as pre-tilt, tilt, and post-tilt. A closed-loop model of interactions was estimated for each interval using *CardioRVAR*, and the BRS transfer functions, as well as the feedforward transfer functions, were isolated from each model. To ensure the reliability of the measurements, the spectral squared coherence, also computed by *CardioRVAR*, was considered alongside each transfer function, and only those frequencies whose spectral coherence was higher than 0.5 n.u. were used in the computations. For comparison purposes with the coherence-thresholding method, estimates were also recovered using the other estimation strategies available in *CardioRVAR*.

The obtained closed-loop BRS indices from these subjects are reported in Table 1. Subject A exhibits a weak response during tilt (Figure 8A), with a rather bradycardic behavior. As can be seen, weak changes in HR rate and SBP are found in this subject during each interval of the HUT test. However, changes in the baroreflex components are highly evident. The HF component of the BRS is highly predominant during the pre-tilt and post-tilt periods, which could be correlated and would explain the bradycardic basal state of this subject. In fact, the coherence levels are below 0.5 n.u. for the LF band during post-tilt, making the BRS estimate unreliable according to the coherence threshold criterion for said band. The Gaussian-averaging estimation and the estimate at maximum coherence methods are able to give individual estimates for these intervals. However, during post-tilt, they differ in their magnitudes for the LF band: after visual inspection, it was noticed that around 0.06 Hz, where the maximum coherence level was located, the gain of the transfer function dropped, giving a low estimate for this coherence level, while, at frequencies around 0.1 Hz, the gains computed were of much higher magnitude. Therefore, we attribute this finding to the fact that the maximum coherence level, at which the gain dropped, was not located near the center of the frequency band, a region that is better explored by the Gaussian-averaging method [35], which in this case had gains of greater magnitude. During tilt, the response at the HF band is reduced, but the activity of the LF component is still not predominant.

Subject B, who participated in a similar protocol, exhibited, however, an intense response during tilt (Figure 8B). This subject had recovered from COVID-19 infection and was suffering from long-COVID-induced POTS. The data were analyzed by the software and its corresponding results are also reported as with subject A in Table 1. As can be seen, the strong HR response during tilt is accompanied by a strong decrease in the HF baroreflex component according to the non-thresholding estimation methods, and a decrease in the coherence levels below the threshold, thus not allowing a correct estimation for this band according to the coherence criterion method. The LF component also decreases, but tends to be predominant during tilt according to the non-thresholding estimation methods. Both components increase again after HUT.

### 3.2. Comparison between Normotensive and Hypertensive Subjects

Table 2 summarizes the obtained results from seven hypertensive and five normotensive subjects from the analysis of recordings from HUT tests. Two methods for the estimation of BRS were used in the analyses: a weighted-averaged BRS without coherence thresholding, as in McLoone and Ringwood’s work [35], and BRS indices obtained at the maximum coherence level for each frequency band. Significant differences are found for both bands in the supine position, and for the LF band during HUT; however, disagreement between the estimation methods is detected at the LF band in the supine position.

No significant differences were found for the HF band during HUT. In certain distributions, normality was rejected through the Shapiro–Wilk test due to the presence of extreme values in said distributions. These extreme values were mainly attributed to the previously discussed subject A, who was included among the normotensive subjects and whose characteristics were previously discussed. One should note that both strategies to compute the BRS tend to share the same statistical conclusions, except for the LF band during supine rest, for which the estimates produced by windowing and averaging through the frequency band were not considered to have a statistically significant difference, whereas the estimates produced by considering the BRS at its maximum coherence level did, although a tendency was evidenced regarding the weighted-averaged estimates. The significant differences found between normotensive and hypertensive subjects correlate with findings reported in the literature, which show that hypertensive patients tend to have lower BRS levels when compared to normotensive ones [40].

### 3.3. EUROBAVAR Analysis Results

We also tested the tool on the open-access EUROBAVAR data set, as done in previous works in the literature [34,35,38,39], including the evaluation of our previous contribution [16]. Preliminary results from the closed-loop BRS assessment from the EUROBAVAR data have been previously presented at a Spanish Society of Physiological Sciences meeting [41]. A set of individual results obtained through the maximum coherence method from each subject is displayed in Figure 9 and Figure 10, divided according to the reported A and B series of the data set [34,38]. For each of these subjects, after uploading their respective recordings into *CardioRVARapp* and preprocessing them, a bivariate VAR model was estimated to model each arm of the closed loop and was transformed into the frequency domain, while incorporating an instantaneous transfer path from BPV to HRV. Then, *CardioRVARapp*-reported estimates from these models were then computed using the maximum coherence strategy (Figure 9 and Figure 10) or McLoone and Ringwood’s weighting strategy.

It should be noted that although McLoone and Ringwood originally applied their weighting method on the EUROBAVAR data set, the strategy was applied to models derived from a direct approach strategy proposed by both authors, analyzing the closed-loop baroreflex system as if it were an open one [35]. *CardioRVAR*, however, aims to follow a different approach, by considering both arms of the closed-loop system in its computations and combining the computed models with the previously described individual estimation strategies. By applying its entire workflow to these data, and through the comparisons of the obtained results with the available works on this data set, the performance of *CardioRVAR* in this setting can be validated.

For comparison purposes, estimates computed through the coherence-thresholding strategy for the A series (Figure 9C,D) and the B series (Figure 10C,D) are also shown together with the ones computed through the maximum coherence strategy. It can be noticed that the coherence-thresholding strategy is not able to produce estimates from certain subjects, particularly from the B series: regarding the impaired subjects [34,35,38], both estimates could be recovered from one of these subjects in the supine position, while estimates from the other one could not be recovered from any band. In the standing position, only estimates for the HF band could be computed from these subjects. This not only affected the impaired subjects, but also some of the non-impaired ones (Figure 9B and Figure 10B,D). The maximum coherence strategy, however, is able to produce estimates for both bands of interest for every subject in the data set, which tend to be similar to the ones produced through the coherence-thresholding strategy. The estimates computed by *CardioRVAR* correlate with findings described in the literature: the baroreflex impairment tends to be better assessed qualitatively in the standing position [35], allowing the identification of the two impaired baroreflex subjects [35]. There is also an extreme baroreflex component in subject 13 from the B series, visible in the supine position (Figure 10A,B), expected according to the literature [38], which shares characteristics with the previously discussed subject A. Subject A008 also tends to give high BRS values for the HF band when compared to the rest of the A series, which is similar to what was observed by McLoone and Ringwood through their methodology [35].

Figure 11A shows changes in baroreflex sensitivity estimates assessed by the *CardioRVAR* after applying McLoone and Ringwood’s weighting method to the computed VAR models for all patients, excluding subjects 5 and 10 from the B series due to their baroreflex impairment as in [16], showing significant differences between the supine and standing positions for the HF band (*p* < 0.001) and the LF band (*p* < 0.001), with supine-to-standing rations of (mean ± S.D.) 3.21 ± 1.44 and 1.98 ± 0.88 for each band, respectively. Figure 11B shows the same estimates after applying the maximum coherence strategy, obtaining supine-to-standing ratios of 2.89 ± 1.17 and 1.91 ± 1.21 for the same bands, respectively. Significant differences between the supine and standing positions for the HF band (*p* < 0.001) and the LF band (*p* < 0.01) were also observed again. These results are summarized in Table 3, which also shows estimates obtained from open-loop strategies, classified as types I and II according to Barbieri et al. [5]. While these estimates were computed through the Gaussian-weighting and the maximum coherence procedures, they share several properties already described for open- vs. closed-loop estimates [5]: type I open-loop estimates tend to be higher than type II estimates, and closed-loop estimates tend to be smaller than open-loop ones, as shown in Table 3. The overall significant differences found between the estimates also correlate with other results reported in the literature [16,35,39], validating the ability of *CardioRVAR* to identify these features.

An analysis of causal coherence (Figure 12, Table 4) in both positions was performed on the same subjects, which revealed, as briefly reported in the Spanish Society of Physiological Sciences meeting [41], that the IBI-to-SBP coherence was significantly higher than the SBP-to-IBI one for the LF band in the supine position (*p* < 0.001), whereas, in the standing position, both coherence flows were statistically equally predominant. For the HF band, no significant changes in causal coherence were observed. The changes observed in the causal coherence in the LF band are important in the context of BRS, as they follow a similar pattern as reported in the literature, which is indicative of baroreflex mechanisms taking place in the analyzed system [6].

### 3.4. Comparison with Other Works

A comparison with other works published in the literature is provided. As explained before, the preprocessing steps used in *CardioRVAR* to validate the produced VAR models are based on the preprocessing workflow proposed by Barlett and Seth in their works [13,14,15]. In *CardioRVAR*’s workflow, the discrete wavelet transform is used in the preprocessing steps of the signals, but not in the analyses per se. This differs from a recently published package from our team, *BaroWavelet* [16], in which wavelet transforms are used as the core of the analysis of the BRS. In this sense, *BaroWavelet* offers particular utility regarding BRS analysis, which is the application of the multiresolution approach of the wavelet transform for a time-frequency assessment of the BRS, aiming for the isolation of both time and frequency components at the same time while mitigating the possible loss of information in these components [16]. Here, *CardioRVAR* can only provide analysis results in the frequency domain. However, it should be highlighted that the main focus of *CardioRVAR* lies in the obtention of closed-loop interaction models of the cardiovascular dynamics. Thus, the simultaneous use of both tools could probably allow the obtention of a more complete overview of the cardiovascular profile, through the combination of both methodological strategies offered by each tool.

*CardioRVAR* also allows users to choose amongst several different strategies to compute estimates from the frequency-domain models. The results depicted in Table 1 as well as Figure 9 and Figure 10 serve as examples of why several estimation methodologies that allow the computation of as many estimates as possible while maintaining the reliability of said estimates at the same time must be included in the software, which is in line with the opinions of several authors [34,35]: estimates from certain subjects may be compromised due to the application of harsh validation methods, such as the application of a coherence threshold (Figure 10B,D), in which case a softer validation strategy could be appropriate to compute said estimates (Figure 10A,C). Thus, *CardioRVAR* includes several coherence-independent mechanisms described in the literature and reviewed in this work to produce these estimates, apart from the classical coherence-thresholding method.

The package *grangers*, also developed for R, offers a very useful set of tools to analyze causal connectivity between sets of signals, which represents another way to demonstrate the presence of causal baroreflex interactions among the measured recordings. However, it does not consider a detailed analysis of the transfer functions that define this system of interactions. Its usage in combination with *CardioRVAR* is nevertheless highly suggested, as *grangers* can also generate frequency-domain significance testing results for the evaluation of the present causal interactions [17].

Although the package *RHRV* is not specifically designed to analyze BRS, its contribution to the R environment for the analysis of HRV is outstanding. The package *BaroWavelet* was in fact designed to work together with the *RHRV* algorithms to produce some of its BRS estimates [16]. While this is not the case for *CardioRVAR*, we strongly encourage its use in combination with *RHRV* in order to generate a complete assessment of the cardiovascular profile.

Finally, it should be noted that *CardioRVAR*, when compared with other tools for closed-loop BRS assessment, is also accompanied by a graphical user interface to simplify the analysis for users. In this way, users that are not used to the mechanics of the R environment can conduct the same analyses with the help of the interface. This strategy was also followed for the *BaroWavelet* package [16], for the same reasons.

## 4. Conclusions

In this work, a new R package and *Shiny* application capable of evaluating closed-loop cardiovascular interactions have been described. A review of the methodological aspects of the tool is provided, and several demonstrations of analyses performed by the tool are given. These analyses tackled the characterization of postural changes and baroreflex impairment, the identification of decreases in baroreflex sensitivity caused by hypertension, and a descriptive analysis in the context of COVID-19-induced POTS. With these results, the potential clinical and scientific applications of this R-based package, *CardioRVAR*, have been highlighted. This package therefore serves as a contribution to the R statistical environment, allowing the analysis of closed-loop cardiovascular dynamics through the reviewed algorithms.

The tool can be accessed through two public GitHub repositories, for both the main package, through the URL https://github.com/CIMES-USNA-UMA/CardioRVAR, and its graphical user interface through https://github.com/CIMES-USNA-UMA/CardioRVARapp. Thanks to the nature of this package, new updates can be introduced in the future to further improve its structure and data analysis algorithms, enabling the possible continuation of this work.

## Figures and Tables

**Figure 1 biology-12-01438-f001:**
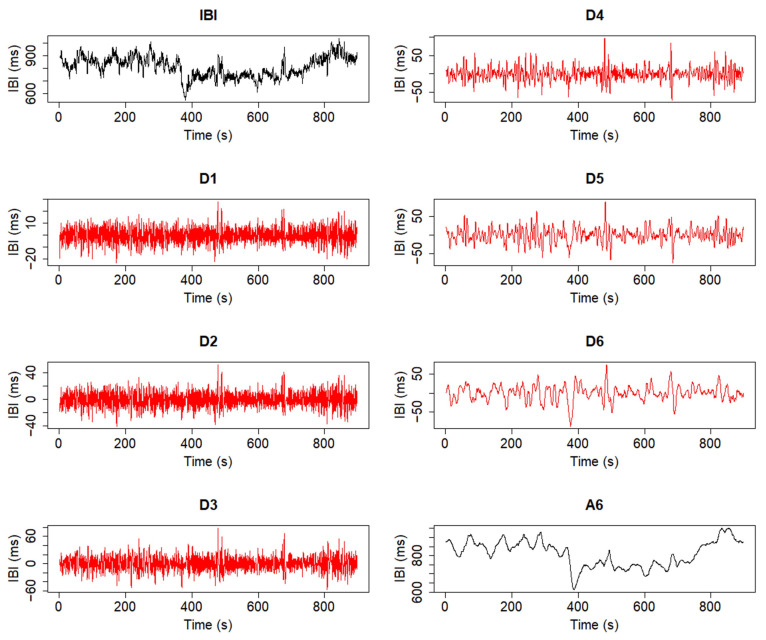
Trend isolation process through a six-level decomposition with the MODWT. The coefficients shown belong to the original signal and its smoothed representation through approximation coefficients (black), as well as to each decomposition level from D1 to D6, which hold the detail coefficients obtained from the analysis (red).

**Figure 2 biology-12-01438-f002:**
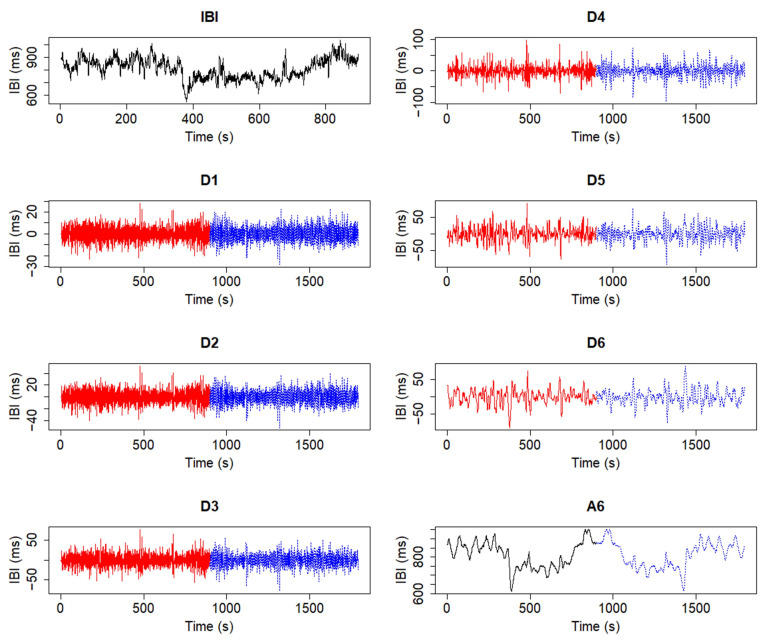
Trend isolation process through the MODWT with an additional border handling strategy applied to the transform. The handling strategy extends the original IBI signal and creates an additional number of coefficients (blue).

**Figure 3 biology-12-01438-f003:**
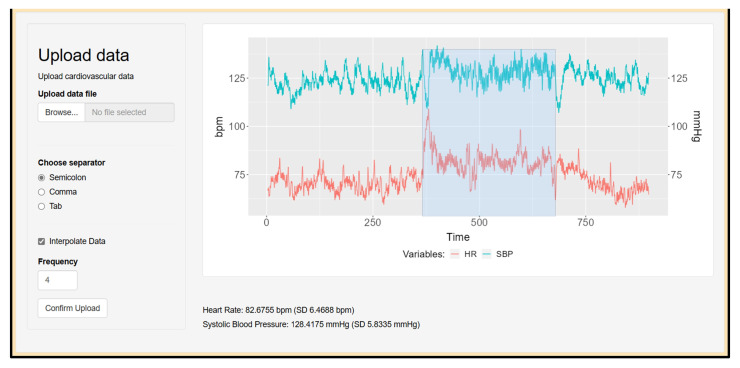
Interface to upload and window cardiovascular data. A csv file containing the necessary data is uploaded and interpolated up to a sample frequency of 4 Hz. A time window is selected, and mean HR and SBP values from this window are reported. The data were obtained from a subject during a head-up tilt session.

**Figure 4 biology-12-01438-f004:**
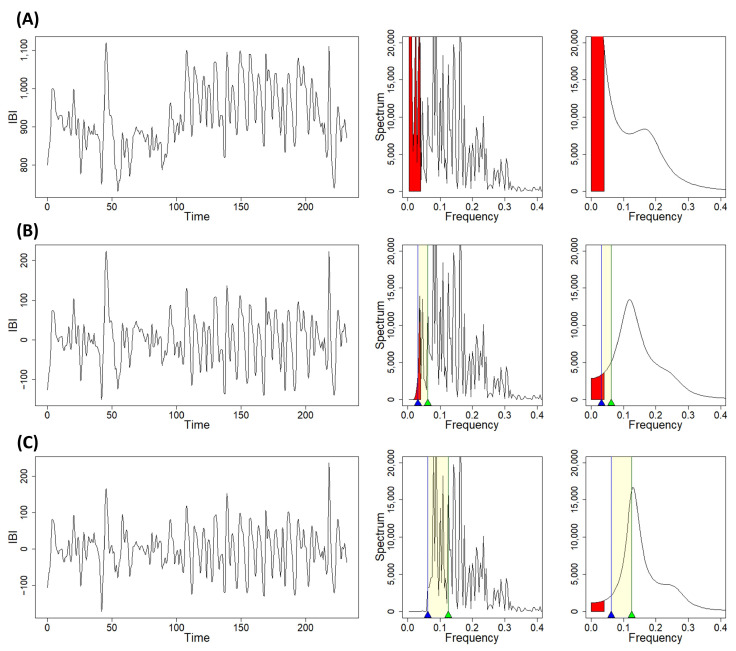
Time-domain and frequency-domain changes in HRV due to the preprocessing algorithms available in *CardioRVAR*. (**A**) Non-detrended signal, with a sample frequency of 4 Hz, and its frequency-domain non-parametric and parametric spectra, with a significant very-low-frequency component. (**B**) Detrended signal after selecting a reference frequency of 0.04 Hz (cutoff: 0.03125 Hz, blue mark), accompanied by its non-parametric and parametric spectra, showing a mitigated very-low-frequency component. (**C**) Detrended signal after selecting a reference frequency of 0.07 Hz (cutoff: 0.0625 Hz, blue mark), with spectral densities showing a more mitigated very-low-frequency component, while also affecting part of the LF component. Yellow areas indicate ranges of possible reference frequencies associated with the last decomposition level, used to identify the cutoff frequency (blue marks) in each case. Green marks indicate cutoff frequencies (0.0625 Hz, 0.125 Hz) associated with the previous decomposition level. Red areas indicate frequency components below 0.04 Hz.

**Figure 5 biology-12-01438-f005:**
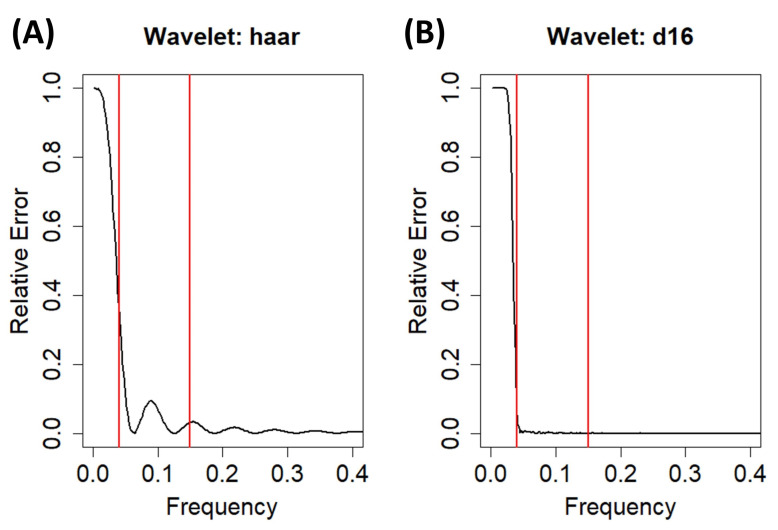
Relative errors computed for each frequency between the frequency-domain non-parametric spectra of a raw IBI signal and its detrended version using the MODWT-based detrending algorithm with (**A**) the Haar wavelet and (**B**) the Daubechies 8 wavelet. Vertical lines indicate frequencies 0.04 Hz and 0.15 Hz.

**Figure 6 biology-12-01438-f006:**
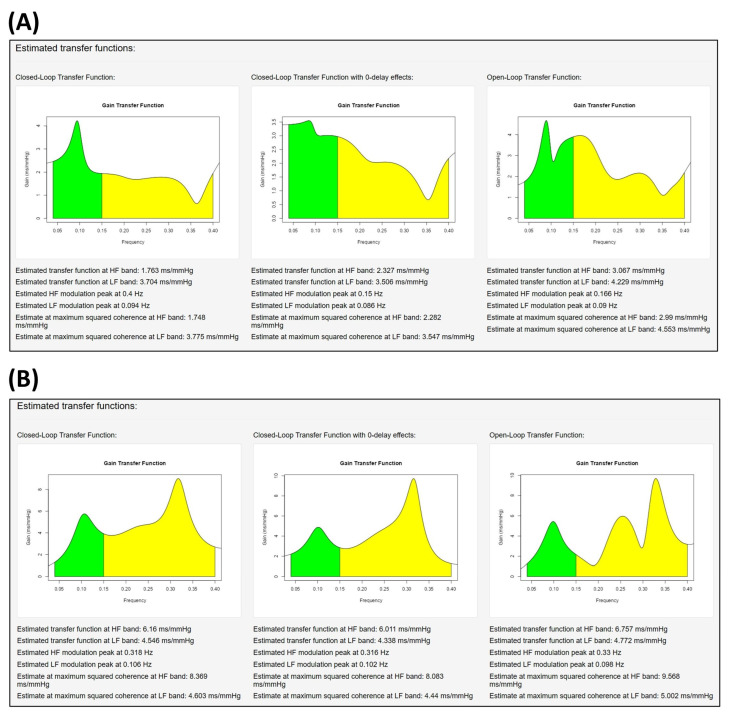
Transfer functions reported by the software from patient during (**A**) head-up tilt and (**B**) post-tilt recovery. From left to right: closed-loop transfer function without zero-lagged interactions, closed-loop transfer function with zero-lagged interactions, and open-loop transfer function. Color code: LF band (green), HF band (yellow).

**Figure 7 biology-12-01438-f007:**
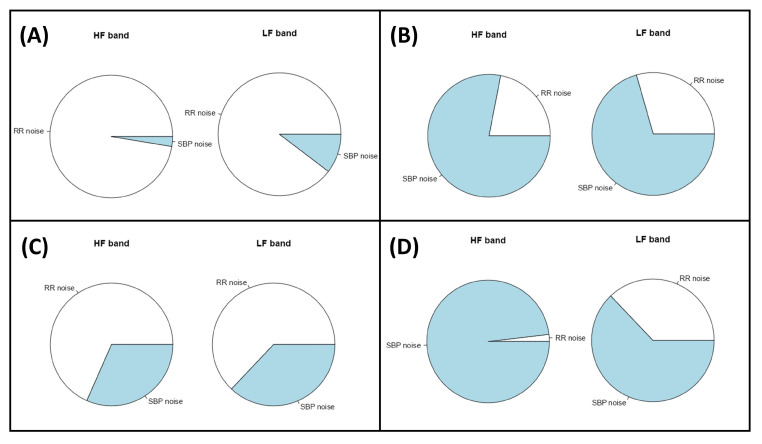
Percentage of noise source contribution computed from a patient to the variability of (**A**) IBI before head-up tilt, (**B**) SBP before head-up tilt, (**C**) IBI during head-up tilt, and (**D**) SBP during head-up tilt. Color code: IBI noise (white), SBP noise (blue).

**Figure 8 biology-12-01438-f008:**
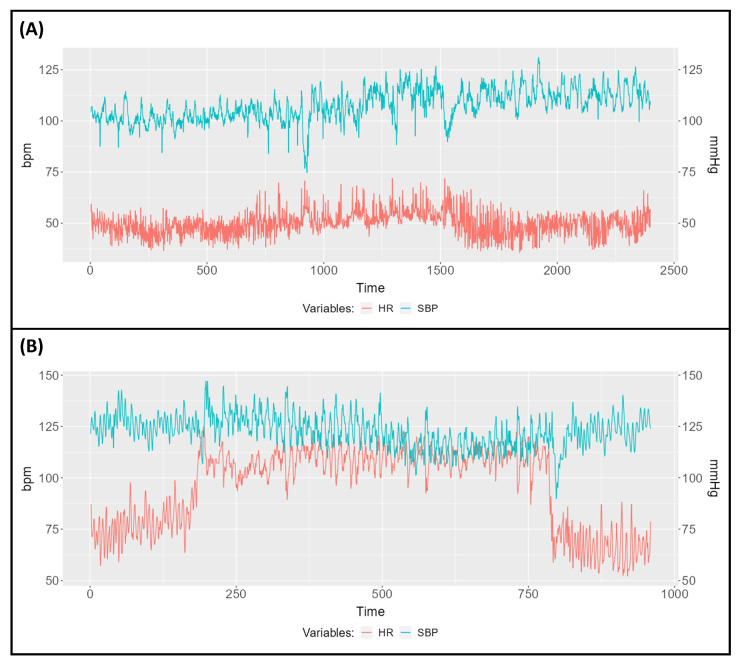
Heart rate (red) and systolic blood pressure (blue) recordings obtained from (**A**) subject A, a healthy patient who exhibited a weak response during head-up tilt, and (**B**) subject B, who suffered from Postural Orthostatic Tachycardia Syndrome induced by long-COVID.

**Figure 9 biology-12-01438-f009:**
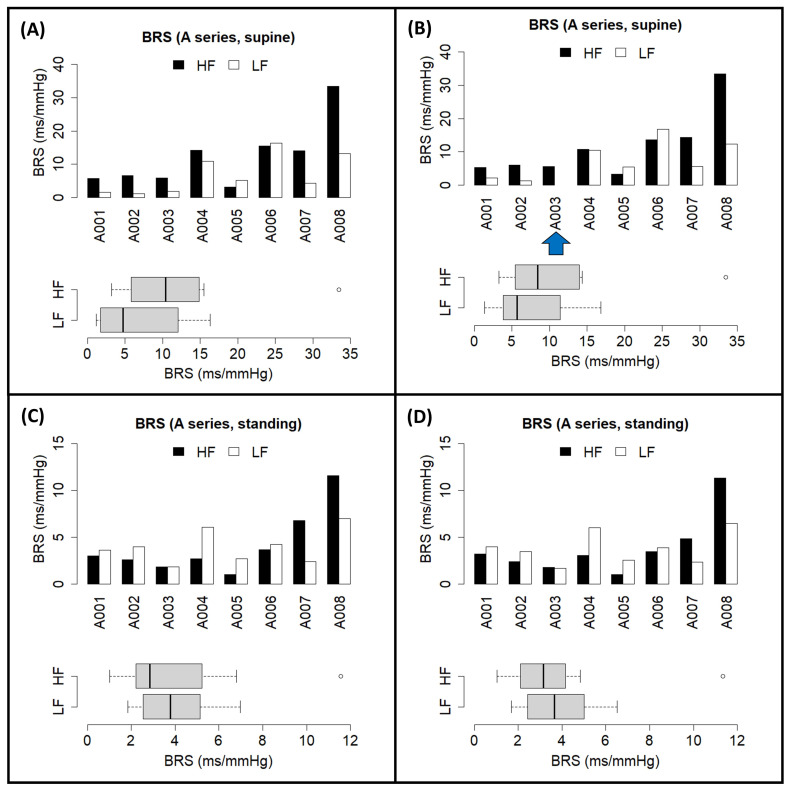
Individual closed-loop BRS estimates computed through *CardioRVARapp* from the EUROBAVAR A series of subjects, and their distributions, obtained in (**A**) supine position, through the maximum coherence strategy; (**B**) supine position, through the coherence-thresholding strategy; (**C**) standing position, through the maximum coherence strategy; and (**D**) standing position, through the coherence-thresholding strategy. Blue arrows indicate subjects with missing estimates due to the coherence-thresholding strategy. Color code: HF band (black), LF band (white).

**Figure 10 biology-12-01438-f010:**
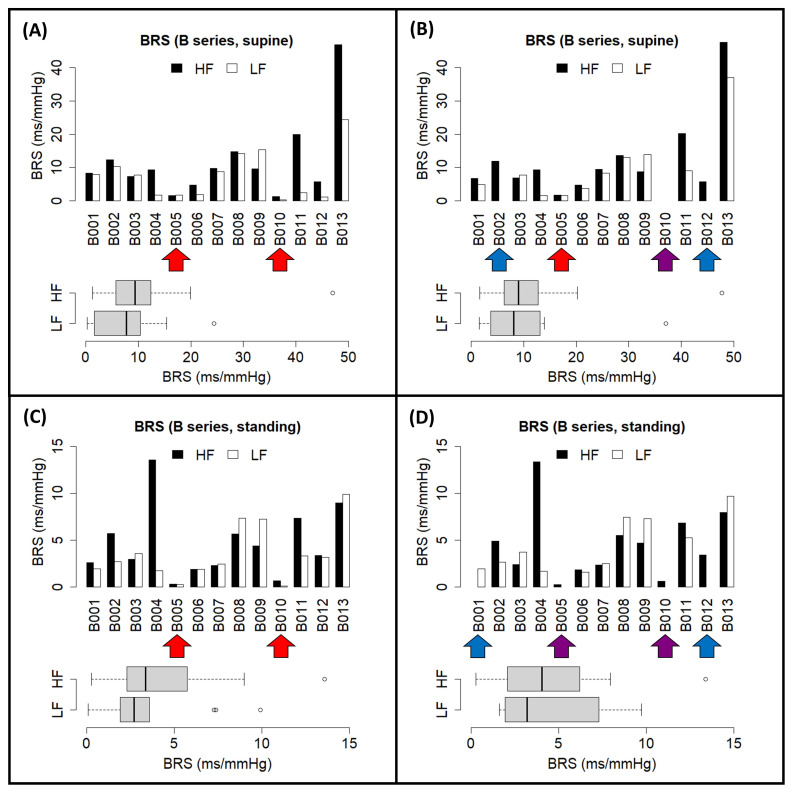
Individual closed-loop BRS estimates computed through *CardioRVARapp* from the EUROBAVAR B series of subjects, and their distributions, obtained in (**A**) supine position, through the maximum coherence strategy; (**B**) supine position, through the coherence-thresholding strategy; (**C**) standing position, through the maximum coherence strategy; and (**D**) standing position, through the coherence-thresholding strategy. Red arrows indicate baroreflex-impaired subjects. Blue arrows indicate non-impaired subjects with missing estimates due to the coherence-thresholding strategy. Purple arrows indicate baroreflex-impaired subjects with missing estimates due to the coherence-thresholding strategy. Color code: HF band (black), LF band (white).

**Figure 11 biology-12-01438-f011:**
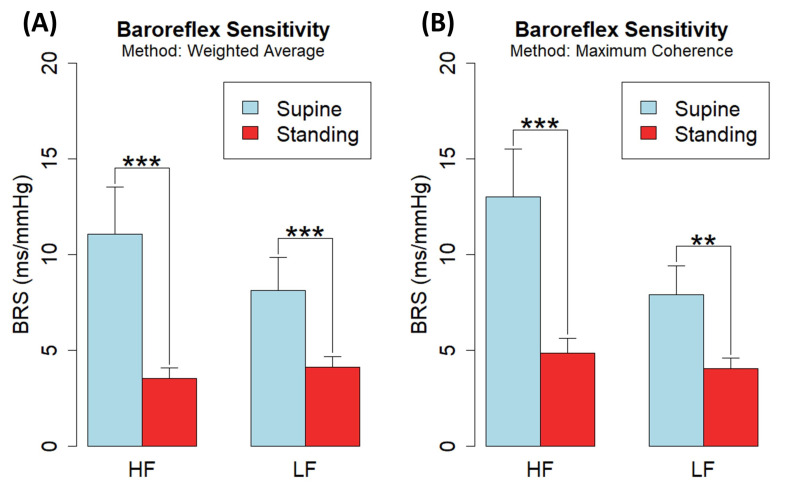
Baroreflex sensitivity changes in closed-loop estimates computed by *CardioRVARapp* from EUROBAVAR during supine and standing positions in HF and LF bands, excluding the two impaired ones (*n* = 19), through (**A**) Gaussian-weighting strategy and (**B**) maximum coherence strategy. Significance: *p* < 0.01 (**), *p* < 0.001 (***).

**Figure 12 biology-12-01438-f012:**
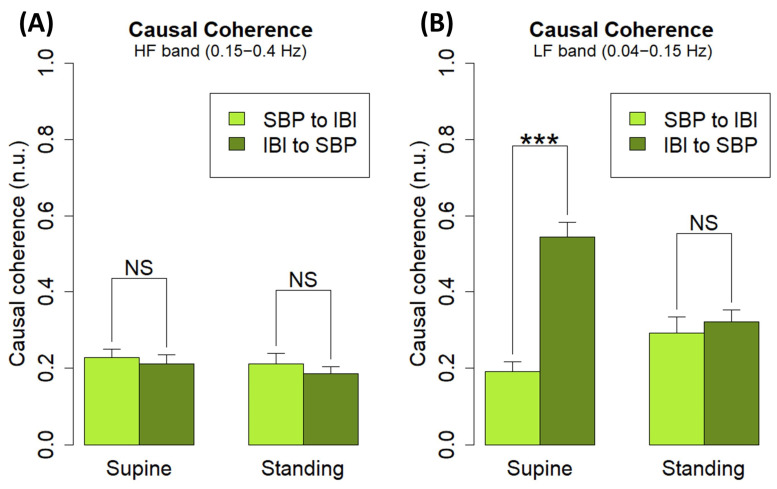
Causal coherence flows from (**A**) HF band from IBI to SBP (dark green) and vice versa (light green) at both supine (left) and standing (right) positions, and (**B**) HF band from IBI to SBP (dark green) and vice versa (light green) at both supine (left) and standing (right) positions. The two impaired subjects were excluded from the analysis (*n* = 19). Significance: *p* < 0.001 (***).

**Table 1 biology-12-01438-t001:** Estimates from subjects A and B during the different periods of the head-up tilt test returned by *CardioRVAR*, using its graphical interface. The displayed results were obtained through the following estimation methods and criteria: coherence threshold (CT), weighted average (WA), and maximum coherence (MC).

	Subject A			Subject B		
Variable	Pre-Tilt Interval	Tilt Interval	Post-Tilt Interval	Pre-Tilt Interval	Tilt Interval	Post-Tilt Interval
HR (bpm)	48.27	54.03	49.42	78.42	109.74	69.75
SBP (mmHg)	102.12	109.15	111.65	125.84	121.62	120.95
CT-HF α_c_ (ms/mmHg)	26.54	11.57	35.90	13.30	N/A ^†^	13.60
WA-HF α_c_ (ms/mmHg)	24.13	9.05	31.56	9.54	1.76	12.81
MC-HF α_c_ (ms/mmHg)	27.34	12.50	36.33	13.10	1.78	16.33
CT-LF α_c_ (ms/mmHg)	6.95	1.74	N/A ^†^	11.78	3.98	17.88
WA-LF α_c_ (ms/mmHg)	7.33	4.59	17.41	11.28	3.81	17.78
MC-LF α_c_ (ms/mmHg)	7.02	1.73	1.63	12.25	4.11	16.77

^†^ Squared coherence was below 0.5 n.u.

**Table 2 biology-12-01438-t002:** Comparison of estimates obtained from the tool between normotensive and hypertensive subjects using two strategies for the computation of the BRS.

Position	Band	Estimate Type	Normotensive (*n* = 5)	Hypertensive (*n* = 7)	*p* Value
Supine rest	HF	Weighted-averaged	9.02 ± 3.88	2.03 ± 0.45	***p* < 0.01**
		Estimate at maximum coherence	10.99 ± 4.14	3.10 ± 0.75	***p* < 0.05**
	LF	Weighted-averaged	5.94 ± 1.38	2.25 ± 0.39	*p* = 0.054
		Estimate at maximum coherence	6.19 ± 1.32	1.69 ± 0.37	***p* < 0.05**
Tilt	HF	Weighted-averaged	4.34 ± 1.39	1.27 ± 0.29	*p* = 0.091
		Estimate at maximum coherence	5.01 ± 1.95	1.46 ± 0.21	*p* = 0.143
	LF	Weighted-averaged	4.90 ± 0.64	2.06 ± 0.22	***p* < 0.01**
		Estimate at maximum coherence	4.69 ± 0.96	1.66 ± 0.23	***p* < 0.05**

Data are presented as means ± standard errors of the mean (SEM). Normality was first tested with the Shapiro–Wilk test. Significance was computed with the unpaired *t*-test or Wilcoxon test. Significant *p* values (*p* < 0.05) are shown in bold.

**Table 3 biology-12-01438-t003:** Supine and standing position BRS estimates computed by *CardioRVARapp* through Gaussian-weighting and maximum coherence strategies from EUROBAVAR subjects, excluding the two impaired ones (*n* = 19).

		Closed-Loop			Open-Loop (Type II)			Open-Loop (Type I)		
Band	Method	Supine (ms/mmHg)	Standing (ms/mmHg)	*p* Value	Supine (ms/mmHg)	Standing (ms/mmHg)	*p* Value	Supine (ms/mmHg)	Standing (ms/mmHg)	*p* Value
HF	Weighted average	11.06 ± 2.46	3.54 ± 0.54	***p* < 0.001**	13.82 ± 2.92	5.03 ± 0.81	***p* < 0.001**	20.79 ± 3.74	7.82 ± 1.27	***p* < 0.001**
	Maximum coherence	13.03 ± 2.47	4.84 ± 0.79	***p* < 0.001**	16.07 ± 2.68	6.40 ± 1.07	***p* < 0.001**	17.40 ± 2.92	7.51 ± 1.38	***p* < 0.001**
LF	Weighted average	8.12 ± 1.72	4.12 ± 0.55	***p* < 0.001**	9.23 ± 2.25	5.12 ± 0.77	***p* < 0.001**	12.72 ± 2.75	7.12 ± 0.91	***p* < 0.001**
	Maximum coherence	7.92 ± 1.50	4.06 ± 0.54	***p* < 0.01**	10.48 ± 2.04	5.42 ± 0.68	***p* < 0.01**	12.43 ± 2.25	6.41 ± 0.77	***p* < 0.01**

Data are presented as means ± standard errors of the mean (SEM). Normality was first tested with the Shapiro–Wilk test. Significance was computed from *t*-test or Wilcoxon test for paired samples. Significant *p* values (*p* < 0.05) are shown in bold.

**Table 4 biology-12-01438-t004:** Supine and standing position causal coherence estimates computed by *CardioRVARapp* from EUROBAVAR subjects, excluding the two impaired ones (*n* = 19).

Position	Band	${{Coh}^{2}}_{SBP\to IBI}$ (n.u.)	${{Coh}^{2}}_{IBI\to SBP}$ (n.u.)	*p* Value
Supine	HF	0.23 ± 0.02	0.21 ± 0.02	*p* = 0.644
	LF	0.19 ± 0.03	0.54 ± 0.04	***p* < 0.001**
Standing	HF	0.21 ± 0.03	0.19 ± 0.02	*p* = 0.510
	LF	0.29 ± 0.04	0.32 ± 0.03	*p* = 0.606

Data are presented as means ± standard errors of the mean (SEM). Normality was first tested with the Shapiro–Wilk test. Significance was computed from *t*-test for paired samples. Significant *p* values (*p* < 0.05) are shown in bold.

## Data Availability

The EUROBAVAR data set is available in open-access format. The raw data supporting the conclusions of this article will be made available by the authors, without undue reservation.

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
