# Peer review of "CardioRVAR*: A New R Package and Shiny Application for the Evaluation of Closed-Loop Cardiovascular Interactions"

_biology, 2023, doi:10.3390/biology12111438_

Round 1
Reviewer 1 Report
Comments and Suggestions for Authors
In their manuscript „CardioRVAR: a new R package and Shiny application for the evaluation of closed-loop cardiovascular interactions“ the authors present their novel R package CardioRVAR which was developed to provide researchers and clinicians with an easily-accessible workflow to estimate baroreceptor sensitivity. The authors achieved this by computing a closed-loop interaction model which requires heart rate, blood pressure and time points as input variables. I congratulate the authors on their innovativeness as well as their programming and statistical skills. The exemplified applications presented in their work provide a promising outlook for a better research workflow in the field of cardiovascular research and I hope to see more of their work in the future.
However, I had two major and a few minor issues that I want to bring to the authors’ notice. As a disclaimer, I am a medical doctor with advanced skills in application of multivariate data integration workflows using R. Thus, in my review I mainly focused on (i) user-friendliness of the CardioRVAR workflow presented in the manuscript, (ii) clinical applicability and (iii) the wording of the text.
Major issues:
i. I experienced difficulties in trying out the workflow, as there was no example data set implemented in the package. After looking through the authors’ GitHub repository, I found the “Cardiovascular” data set, which I then used to run the models presented in the manuscript’s CardioRVAR workflow. However, I did not find any instructions on how to correctly visualize these models and the caveats that should be considered for respective plots/models. I would kindly ask the authors to complement the workflow with an example vignette and toy data frame that serves as an instruction. This would greatly improve the applicability of the methodology.
ii. Regarding the CardioRVARapp, I was unable to upload the data in a suitable format. I tried exporting the authors’ “Cardiovascular” dataset into a .csv and a .txt file, each both as (a) list format (I tried rio::export_list() and utils::capture.output()) and as (b) data frame (using rio::export()).
Regarding (b), I was able to run the model for SBP but not for HR (it showed a hyperbolic graph).
I am operating on R Version 4.2.2 and RStudio Version 2023.6.0.421 and use the Microsoft Edge Browser. For reference, here is the structure of the data frame curated from “Cardiovascular” (first 6 rows), as indicated in lines 173 and 180-181:
Time IBI SBP
0 800 130.8288574
0.839999999999918 840 132.2021484
1.69000000000005 850 129.8217773
2.6099999999999 920 131.7443848
3.6099999999999 1000 132.6599121
4.6099999999999 1000 128.2653809
Although the authors state in line 182 that they used the output of the ACQKnowledge software, I was unable to find an example of such an output online. If I might suggest a practical solution to this problem, it would be helpful if an example csv or txt file were to be made available, so users can more easily adjust their data structure to the application’s needs. As a reference, the MetaboAnalyst workflow offers a toy data set and has an option to run their app with said data to demonstrate an “ideal” workflow (https://www.metaboanalyst.ca/MetaboAnalyst/upload/EnrichUploadView.xhtml).
Minor issues:
General comments:
i. As I was trying to get my input data into the desired structure, I noticed that the CardioRVARapp self-terminated when encountering an error. Thus, I had to repeatedly restart the shiny application using the suggested command shiny::runGitHub("CardioRVARapp", "CIMES-USNA-UMA", subdir = "inst/app", launch.browser = TRUE). Since this proved to be a rather tedious process, I wanted to ask the authors whether a negative feedback loop allowing to stay in CardioRVARapp would be feasible.
ii. In lines 37-41 the authors describe the output of HRV. As far as I am aware, only heart rate recordings are needed to calculate HRV, not blood pressure measurements. Perhaps “BPV” was omitted in line 37?
iii. The annotations in figure 1 are rather difficult to read. I am aware that this might be due to the file compression for the submission processes. However, if it is possible to increase the annotation font size a little bit, I believe it would increase the readability of figure 1.
iv. Why do the authors use the auxiliary verb “would” in line 81 when describing equations 5 and 6?
v. In lines 145, 440 and 441 the authors refer to two GitHub repository with their respective links – however, when accessing these links, I get redirected to the same GitHub page. Is that intended?
vi. The authors present CardioRVAR’s workflow in an interesting and engaging manner using three different examples. How do the models obtained through CardioRVAR differ in performance and data presentation/accessability when compared to other workflows currently available for public use (e.g. the RHRVpackage or Barbieri et al.’s equation, as mentioned in the introduction)?
vii. As a stylistic suggestion to enhance readability, would it be possible to make any packages mentioned in the manuscript in italic?
Semantics-/spelling-related comments:
i. Line 41: As the authors refer to both LF and HF, I believe the plural of “this” is necessary.
ii. Line 45: Do the authors mean “… the magnitude that correlates with both BPV and HRV…”?
iii. Line 47: Please change “measurments” to “measurements”.
iv. Line 68: Please change “describe” to “describes”.
v. Line 70: As the authors use the subjunctive mood, I believe the correct way to phrase this sentence is “If this model were to be …”
vi. Line 73: Please add a comma after “therefore”.
vii. Line 77: To increase readability, I suggest the following changes to the sentence: “… , and, thus, of the whole model, as presented in matrix
viii. Line 99: To increase readability, please insert a comma before “depending”.
ix. Line 100: I believe the readability of this section could be improved by restructuring the sentence to start with “Not only …” as this fits well with the thus far excellent flow of the methods section of the manuscript. However, as this merely reflects personal preference, I would like to leave the implementation of this suggestion at the discretion of the authors.
x. Line 216: Do the authors refer with “one” to the user? If so, please clarify the sentence to improve readability. Furthermore, please change “consider” to “considers” if referring to the user in third person singular.
xi. Line 250: Did the authors mean “closed-loop model”?
xii. Line 318: Please change “last” to “final”. Also, please restructure the sentence as I cannot follow what “produced by the tool” refers to.
xiii. Line 324: Please change “hendle then” to “handle them”.
xiv. Line 327: As 3.1. refers to the comparison between two subjects, I believe it is clearer to state “We offer a practical example …”.
Taken together, I see a great potential in the authors’ work and look forward to seeing the updated version of their manuscript. I hope my comments will aid the future utilization of the methodology proposed in this manuscript.
Comments on the Quality of English Language
Overall, the manuscript was easy to follow and the authors managed to establish a well-structured flow. In my opinion, the English level is well-suitable for publication in a peer-reviewed journal. I invite the authors to incorporate the suggestions stated in the semantics-/spelling-related comments above to correct any residual writing mistakes.
Reviewer 2 Report
Comments and Suggestions for Authors
The study sounds interesting and needful. However, few issues need to be addressed.
1. The manuscript has to be grammatically checked.
2. Few lines of Abstract and Introduction seems to be discontinuous.
3. The authors need to add their novelty inside the manuscript.
4. The manuscript lacks enough background literature. Please ensure the correction before publication.
Comments on the Quality of English LanguageThe entire manuscript should be read once and grammatically corrected.
Reviewer 3 Report
Comments and Suggestions for Authors
The developed library is presented very well. A few minor comments:
1. There is no discussion about whether there are any existing statistical / data analysis tools for cardiovascular research. Is there any public code on this? It would be helpful to add some discussion on that and add some comparisons if needed to highlight the contributions.
2. The manuscripts have good citations for the methods. It would be good to add some references to the application side.
Round 2
Reviewer 1 Report
Comments and Suggestions for Authors
I commend the authors for implementing any suggestions flawlessly and in a highly professional manner. All issues I experienced with the code or the GUI have been resolved and the added chapter regarding comparison to other packages has increased the quality of their manuscript to a great deal.
As a final (optional) suggestion, perhaps adding a comment to the top part of GUI stating that an example analysis can be run by clicking "Confirm Upload" without any prior data upload would be helpful.
I fully support the publication of this manuscript and hope to see further work of the authors in the future.
